# Bacteriophages as Biocontrol Agents in Livestock Food Production

**DOI:** 10.3390/microorganisms10112126

**Published:** 2022-10-27

**Authors:** Logan Gildea, Joseph Atia Ayariga, Boakai K. Robertson

**Affiliations:** 1The Microbiology Program, Department of Biological Sciences, College of Science, Technology, Engineering, and Mathematics (C-STEM), Alabama State University, Montgomery, AL 36106, USA; 2The Hemp Program, Department of Biological Sciences, College of Science, Technology, Engineering, and Mathematics (C-STEM), Alabama State University, Montgomery, AL 36104, USA

**Keywords:** bacteriophages, biocontrol, food production, bacterial resistance

## Abstract

Bacteriophages have been regarded as biocontrol agents that can be used in the food industry. They can be used in various applications, such as pathogen detection and bio-preservation. Their potential to improve the quality of food and prevent foodborne illness is widespread. These bacterial viruses can also be utilized in the preservation of various other food products. The specificity and high sensitivity of bacteriophages when they lyse bacterial targets have been regarded as important factors that contribute to their great potential utility in the food industry. This review will provide an overview of their current and potential applications.

## 1. Introduction

An integral part of human life is the consumption of food products. While it is essential to consume food, it also provides a pathway to infection caused by foodborne pathogens such as *Salmonella* spp., *Escherichia coli* (*E. coli*), *Campylobacter* spp., and *Clostridium* spp. [1,2]. Annually foodborne pathogens account for around 38.4 million illnesses, 486,000 hospitalizations, and 6000 deaths in the United States alone [3]. Even with enhanced technology, thorough quality control, and advanced sterilization methods these pathogens persist within the environment. The ever-evolving nature of food production, due to consumer demand and economic considerations, presents challenges in maintaining food quality and safety. Dependent on the food product, common antimicrobial methods in production are good hygiene and various antimicrobial agents. These practices must be applied in every aspect of the food production process to ensure safety and quality. Regardless of the many preventative measures in place, bacterial communities persist through development of resistance, biofilm formation, and other mechanisms [4,5,6]. This bacterial persistence is why development of new novel biocontrol agents, such as bacteriophages, are crucial for these constantly evolving organisms. This review will provide an overview of the relevant literature and commercial products that utilize bacteriophages as agents of biocontrol in the food industry.

## 2. What Is Biocontrol?

Biocontrol refers to the use of one or more organisms to inhibit or control other organisms. Agents of biocontrol can be divided into two categories: living organisms, such as phages or byproducts of an organism, such as the production of bacteriocins. In general biocontrol encompasses the use of lactic acid bacteria, bacteriocins, endolysins, bacteriophages, and ‘protective’ cultures. Biocontrol relies on the ability of agents to carry out microbial interference. Microbial interference refers to the general nonspecific inhibition or destruction of one microorganism by other members of the same environment.

A major benefit of biocontrol, unlike other microbial control methods, is that it is derived from natural and green sources. Biocontrol products are typically less harsh and generally accepted by the public in the current market [7,8]. These agents are characterized as nontoxic, natural, and can withstand physiological conditions as well as maintain a narrow spectrum of antimicrobial activity [9,10]. While bacteriophages represent an up-and-coming biocontrol technique, some biocontrol techniques are currently already widely used. The bacteriocin nisin was first introduced commercially nearly 60 years ago and has been utilized in numerous food and beverage products such as cheese, dairy, canned vegetables, and pasteurized liquid eggs. [9]. Newer biocontrol agents include endolysins, proteins typically derived from bacteriophages, that function to exogenously lyse bacteria. Endolysins are typically paired with *holins*, a phage protein that perforates the peptidoglycan layer of Gram-positive bacteria to allow for entry into the cytoplasm of the cell. Endolysins have been employed as agents of biocontrol against *Clostridium perfringens*, *Clostridium fallax*, and *Listeria monocytogenes* [11,12,13]. A unique and especially important agent of biocontrol, which is the primary focus of this review, is the bacteriophage. Bacteriophage proteins have been utilized as biocontrol agents in recent history, however the utilization and benefits of utilizing the entire bacteriophage has become an area of interest. Bacteriophages (phages) are abundantly present in nature and phage typing has been utilized to identify phages and their associated bacterial hosts. Research focus since the beginning of the 20th century has been on lytic bacteriophages as biocontrol, but more recently these organisms have gained attention as potential agents against antibiotic resistant bacteria.

### The Role of Bacteriophages in Food Microbiology

Bacteriophages infect bacterial species with high specificity [14,15,16,17,18,19,20,21,22,23]. This high specificity allows bacteriophages to target pathogenic foodborne bacteria while not affecting the natural microbiota of food products or the humans who consume them [24]. Phages are found in every environment suitable for bacterial growth and are already present in all the foods we consume. Research on phages have identified bacteriophages associated with almost every known pathogenic and non-pathogenic bacterium [25]. While phages are one of the most abundant organisms on earth it is important to differentiate which phages are appropriate for use in food microbiology. The broadest classification method of phages is based on their lifestyle which can be either lytic or lysogenic. Lytic phages function to kill bacteria through lyses while lysogenic phages allow the bacteria to persist while infected with the virus (Figure 1). In terms of therapeutics and food microbiology, it is inadvisable to utilize lysogenic phages for bacterial control due to the persistence of the bacteria [26]. Lytic phages are preferred for the treatment of bacteria as they do not allow the bacteria to persist which is the desired goal in food microbiology.

Phage biocontrol is progressing towards acceptance as a novel and green technology that could eventually replace some standard methods of intervention such as chemical antimicrobials [27,28,29]. Unlike common chemical antimicrobial interventions such as alcohols, chloride compounds, and bisphenols; bacteriophages are naturally occurring organisms that are mostly safe for consumption and can be readily used at multiple points in food processing without sacrificing the quality or safety of the product [30,31]. Bacteriophage preparations are either distinguished as mono-phage treatments or phage cocktails. Mono-phage treatments consist of one singular phage type and is used to target a singular known pathogen present in the food product. Whereas phage cocktail preparations are a mixture of multiple phage types and can be utilized for treatment of food products with pathogens that readily develop resistance or are diverse. Phage cocktails are designed primarily as a preventative treatment that can be utilized to ensure that common pathogens are unable to colonize a food product. The acceptance of phage treatment has consistently grown since the FDA granted clearance to ListShield^TM^ a *Listeria monocytogenes*-specific phage cocktail used as a food additive in 2006 [32]. Since this major steppingstone, multiple phage treatments for foodborne pathogens including *Salmonella* spp., *E. coli*, and various other common foodborne bacteria have been accepted by the FDA as safe additives to food products. A major step in the progression of phages as a commonly accepted food additive was the FSIS directive 7120.1. This directive stated that addition of phages to livestock prior to slaughter and food is permitted, which promoted the production of multiple lytic phage additives such as *E. coli* O157:H7-targeted phages, used for the reduction of aerosol contamination from the hides of cattle and P22-*Salmonella*-targeted phages in the use of poultry and fresh meats [33,34].

One of the advantages of bacteriophages is their ability to be incorporated in multiple aspects of food processing including animal therapy, sanitation, biocontrol, and preservation. The versatility of phages promotes them as a sensible option that must be developed to enhance and preserve the safety and quality of food products. Adaption of phage treatment methods would provide not only ease of use but also decrease economic loss associated with spoilage and product contamination [35,36]. Table 1 discusses various advantages and disadvantages of bacteriophage biocontrol.

## 3. Phages as Agents of Animal Therapy

An important aspect of food production is the ability to maintain healthy livestock. While many focus on the harvesting and preparation aspect of food production, animal health is a major contributor to the prevalence of foodborne pathogens [51]. Several pathogens can infect farm animals asymptomatically, which can lead to contamination and public health hazards. Many of these pathogen stem from the increased prevalence of antibiotic-resistance within farm animals that can be contributed to the mass and sometimes inappropriate use of antibiotics [52]. Bacteriophages offer several benefits that are not present in antibiotic intervention (Figure 2). While these pathogens are antibiotic resistant, they are still susceptible to phages [53]. This contributing factor is a major reason why the prospect of phage therapy has gained much interest with the post-antibiotic era quickly approaching. In addition to the functionality of bacteriophages, they possess several beneficial characteristics in terms of treatment. While both antibiotics and bacteriophages can eliminate a bacterial pathogen, bacteriophages have been shown to enhance angiogenesis, protect against injury, promote fat storage, enhance immune response and resistance, as well as promote the biosynthesis of vitamins and amino acids that aid in therapeutic metabolism through preservation of natural microbiota (Figure 2). Preserving the natural microbiota of livestock through bacteriophage treatment results in enhancement and promotion of natural human functions including immune response, proper angiogenesis, and nervous system modification. Antibiotic treatment has been criticized due to its inability to preserve natural microbiota responsible for numerous aspects of health [54]. It has been demonstrated that the natural microbiota provides special benefits to livestock and humans alike [54]. These natural bacteria are involved in biosynthesis of vitamins and amino acids, breakdown of complex food compounds and metabolism of therapeutics, development, and training of the bacterial host’s immune system [55,56,57]. Microbiota has also been demonstrated to inhibit the colonization of the gut of their animal host with pathogenic bacteria [58,59]. It has been revealed also that microbiota modifies the nervous system, encourages fat storage, promote angiogenesis, provide protection for epithelial tissues of its host [59,60,61,62,63]. The use of broad-spectrum antibiotics usually leads to the destruction of the microbiota of the animal or the human host [54,64].

## 4. Phages as Therapeutics in Poultry

An area of livestock production where bacteriophages have shown the most promise is in poultry production. The versatility of phages allows them to be incorporated in numerous aspects of the poultry production process. For this reason, a large majority of phage biocontrol studies have been focused on poultry. The most common bacterial pathogens of concern associated with poultry are *Salmonella* spp., *E. coli*, *Campylobacter* spp., and *Clostridium* spp. [65]. These bacterial pathogens have shown to be increasingly resistant to antibiotic treatments as well as antimicrobial compounds [66]. The European Food Safety Authority (EFSA) reported that roughly 28.6% of *Salmonella* spp. and 34.9% of *E. coli* found in food sources were multidrug resistant [67]. Considering this data, it is crucial to incorporate new and novel methods for the treatment of these pathogens. Literature has examined the efficacy of various phage treatments against several common foodborne pathogens such as *Salmonella* spp., *E. coli*, *Campylobacter* spp., and *Clostridium* spp. outlined in the following subsections (Table 2).

### 4.1. Salmonellosis

*Salmonella* spp. are prevalent pathogens in the realm of foodborne disease, causing disease in a variety of endothermic animals, humans, and leading to significant production losses in livestock [68,82]. *Salmonella* spp. can be divided into host-restricted and non-host-restricted classifications. Host-restricted *Salmonella* spp. are characterized as bacterium that produced severe, typhoid-like illness in a single host. Non-host-restricted *Salmonella* serotypes, such as *S. enteritidis* and *S. typhimurium*, typically result in moderate gastrointestinal infections across a broad range of species and account for most foodborne bacterial infections [83]. Phage therapy and biocontrol primarily focus on non-host-restricted *Salmonella* spp. due to their prevalence and relation to most foodborne *Salmonella* outbreaks.

Phage therapy has been extensively studied in the poultry industry for the control of *Salmonella*. Poultry products account for most of the *Salmonella* outbreaks reported in the United States every year [82] so the development of specific biocontrol methods such as phage therapy are essential for the prevention of *Salmonella*-related illness. Phage treatments in poultry can potentially address several issues of *Salmonella* infections such as gut colonization, horizontal transmission, and prevalence of multi-drug resistant infections [33,68,83]. Several studies have outlined the success of various phage treatments aimed to reduce *Salmonella* colonization within poultry prior to harvest (Table 2). In 2007 Atterbury et al., exhibited the effectiveness of *Salmonella* bacteriophages against *S. enteritidis*, *S. hadar*, and *S. typhimurium* [68]. Through bacteriophage administration in antacid suspensions to broiler chickens with experimental infections it was shown that this treatment effectively reduced cecal *Salmonella* colonization approximately 2.19–2.53 log_10_ CFU/g [68].

Some research discovery has revealed that in the case of phage treatment against *Salmonella* infections, the prevalence and persistence of bacteriophage-insensitive mutants (BIMs) can be reduced through responsible treatment methods [33,68]. Phage therapy’s ability to lyse antibiotic-resistant *Salmonella* spp. is also a major benefit in terms of phage incorporation into food production.

### 4.2. Colibacillosis

One of the most prominent pathogens responsible for food contamination and foodborne disease is *E. coli*. A Gram-negative, anaerobic, rod-shaped, coliform bacterium, *E. coli* is responsible for the majority of reported foodborne illnesses and several significant outbreaks in the United States [84]. In terms of food production, *E. coli* is also responsible for considerably high mortality in poultry [85]. This can be attributed to the occurrence of avian respiratory tract infections causing Colibacillosis. Development of alternative preventative methods such as phage treatments is crucial to the reduction of economic loss and potential hazards caused by *E. coli* infections in poultry.

Numerous studies have attempted to progressively introduce and develop phages for the treatment and prevention of *E. coli* infections within poultry (Table 2). The efficacy of phages as a preventative method was confirmed by Huff et al. in 2002 through use of a high phage titer phage cocktail administered through aerosol spray as a therapeutic measure [75]. Phage treatment of *E. coli* challenged chickens reduced the mortality from 50% in the control group to 20% in the treated group [75]. Other studies have utilized aerosol phage treatments as therapeutics for poultry and confirm the reduction of mortality due to phage treatment [86,87]. One of the notable drawbacks of the aerosol treatment of livestock is that it must be administered immediately as studies suggest that efficacy is significantly lowered after 24 h of *E. coli* challenge [76]. To address this efficacy issue, Huff et al., also administered a combination of two phages, DAF6 and SPR02, through direct intermuscular injection into *E. coli* challenged chickens. This treatment method resulted in significantly lower mortality whether the combination was injected immediately or up to 48 h after bacterial challenge [76]. Such a finding sufficiently illustrates the importance of phage administration methods to the successful prevention of foodborne pathogens.

Another potential threat of colibacillosis is the development of meningitis or septicemia. Unlike respiratory infections, in which aerosols can be utilized, these infections require intramuscular or intracranial administration for phage treatment. Research findings on phage has shown that intramuscular or intracranial administration of phage K into *E. coli* challenged chickens resulted in a complete elimination of mortality related to these infections when injected immediately. Even administration following development of clinical signs of infection resulted in a 70% survival rate in phage treated chickens [88]. The ability of phages such as phage K to multiply in blood and penetrate the blood–brain barrier supports the idea that phages could serve as viable treatment methods for acute infections.

### 4.3. Campylobacteriosis

In the United States and European Union (EU) *Campylobacter* is a significant contributor to acute bacterial foodborne disease with 95% of reported diseases being the result of *C. jejuni* [67]. *Campylobacter* has been responsible for six separate outbreaks since 2010 in the United States alone. *C. jejuni* infections are incredibly prevalent and Campylobacteriosis has been linked to Guillian-Barré Syndrome and reactive arthritis [89]. A major obstacle in the prevention of *Campylobacter* contamination in food products is its high adaptability and relatively low infectious dose [90]. Both characteristics significantly lower the efficacy of standard prevention methods such as antibiotics and other antibacterial agents. The UK Food Standards Agency reported increased occurrence of antibiotic resistance in *Campylobacter* spp. against ciprofloxacin, tetracycline, nalidixic acid, streptomycin, and erythromycin [91]. This prevalence stresses the need for new and effective novel prevention and therapeutic agents.

Phage therapy has been studied rather extensively in the treatment of Campylobacteriosis (Table 2). Due to the high adaptability of *Campylobacter* spp. multiple phage therapy or phage cocktails have been suggested as more efficacious treatment methods. *Campylobacter* phages are arranged into three groups based on their structure, genome size, and host receptor [92]. Literature suggests that group I and II *Campylobacter* phages utilize multiple receptors for the infection of host cells making them ideal for treatment of *Campylobacter* infections [93,94,95]. The efficacy of group II and III phages CP14 (III) and CP68 (II) in combination therapy was observed in Vrolix chickens by Hammerl et al. in 2014. Phage cocktail treatment of chickens inoculated with a 10^9^ CFU of *C. jejuni* resulted in a 3 log10 CFU/g reduction of *C. jejuni* colonies within the caeca [77]. It was also concluded that phage cocktail treatment was more efficacious than phage monotherapy using CP14 or CP68. Additionally, the prevalence of resistant *Campylobacter* was significantly reduced through the application of phage cocktail treatment [78]. *Campylobacter* colonization of the caeca in birds does not appear to be very invasive allowing for oral phage administration to be a preferred delivery method. Phage cocktails administered through gavage and incorporated with feed reduced *C. jejuni* and *C. coli* colonization by 2 log10 CFU/g in broiler chickens [78]. The results presented suggest that phage treatments could be a valuable tool for the control of *Campylobacter* spp. in food production.

### 4.4. Clostridiosis

*Clostridium* spp. are considered a pathogen with significant impact due to its association with enteric disease in poultry. These enteric diseases caused by *Clostridium* species, primarily *C. perfringens*, are difficult to diagnose due to clostridial species being present in the gut as a natural microbe [96]. *C. perfringens* is characterized as a Gram-positive, rod shaped, anaerobic, spore-forming bacterium that is one of the leading causes of foodborne disease. Typical diagnostic methods such as alpha-toxin detection and isolation cultures cannot be used for confirmation of infection since they are present within the gut naturally [75]. The most common enteric disease associated with *C. perfringens* in poultry is necrotic enteritis. The poultry industry annually loses 5 to 6 billion USD per year due to the loss caused by necrotic enteritis [97]. This disease is prevalent in broiler chickens but has also been diagnosed in a variety of avian species such as turkeys, waterfowl, and ostriches. The prevention of *C. perfringens* infections in poultry would not only reduce economic loss associated with food production but also reduce the prevalence of related foodborne disease in humans [97].

Phage therapy has been studied as a potential therapeutic in chickens for the reduction of symptoms and disease progression (Table 2). Initial study of *C. perfringens* phages revealed that phage-encoded endolysins target and hydrolyze the peptidoglycan mesh of *C. perfringens* with high efficacy leading to bacterial lysis [80]. The use of purified phage endolysins show promise as a phage-protein therapy to reduce *C. perfringens* colonization [11,12,98,99]. Miller et al., in 2010 determined a five-phage cocktail, termed INT-401, administered orally reduced *C. perfringens* associated mortality by 92% [81]. These results concluded that phage treatments could be a viable agent of biocontrol of *C. perfringens* in poultry. Further study to standardize phage treatments of *Clostridium* spp. is a key to the advancement in treatment of this pathogen.

## 5. Bacteriophages as Biocontrol in Food Production

Bacteriophage biocontrol offers a novel method to target pathogenic bacteria without disturbing the natural microflora of food products. The versatility of phages lends them to application not only as therapeutic agents but also as effective post-harvest treatments for food products [72]. With a comparative efficacy to traditional chemical interventions, novel and natural phages present an environmentally friendly antimicrobial that can be incorporated in various aspects of food production. These phage treatments also do not contain any potentially harmful additives or preservatives and are typically water-based [72]. These attractive aspects of phages have resulted in several phage treatments to be designed and utilized against several notable bacteria including *Salmonella*, *E. coli*, *Clostridium*, *Campylobacter*, and *Listeria*.

*Listeria* spp. are of particular concern in food production. These rod-shaped, Gram-positive, facultative anaerobes are common foodborne pathogens that can induce a wide range of symptoms such as flu-like or gastrointestinal symptoms that can progress to encephalitis or cervical symptoms [40]. The most notable species, *Listeria monocytogenes*, is estimated to be responsible annually for 14,000 infections and 3000 deaths globally [40,100]. While *L. monocytogenes* is relatively rare, its relatively high mortality rate of approximately 20–30% calls for concern [40,100]. This pathogen’s ability to persist and grow at temperatures consistent for refrigeration (2–8 °C) makes it a considerable threat even for prepacked and well refrigerated foods that are typically considered safe to consume. Considering this it is crucial to develop new and novel methodologies for the protection of foods from *L. monocytogenes* contamination.

There are currently several notable GRAS approved *Listeria* phage treatments for utilization in food production. These treatments consist of both mono-phage treatments and phage cocktails utilized specifically against *L. monocytogenes*. A study conducted by Figueirdo et al. in 2017, confirmed that a mono-phage treatment designed against *L. monocytogenes* was more effective in bacteria reduction than nisin or sodium lactate at the storage temperature of 6–8 °C in sliced ham [101]. These significant findings exhibited the effectiveness of bacteriophages in conditions consistent with food refrigeration. Additional studies further confirmed the effectiveness of mono-phage treatments against *L. monocytogenes* in refrigerated ready-to-eat food products [101,102]. Phage-cocktail treatments against *Listeria* spp. have also shown effectiveness against food products contaminated with multiple species including *L. monocytogenes* [103]. Phage biocontrol has shown novel application as a treatment against *Listeria* contaminated food products and for this reason there are numerous *listeria*-phage treatments that have reached approval (Table 3).

In addition to *Listeria*, numerous phage applications are present for other prevalent pathogens such as *Salmonella*, *E. coli*, *Clostridium*, and *Campylobacter* [72]. These phage treatments have been used in applications ranging from decontamination of food preparation surfaces to decontamination of food products directly. Most bacteriophage products in reference to these pathogens are in many cases the same phages used for veterinary intervention in livestock [65,66,67]. Table 3 briefly describes several relevant bacteriophage products approved for usage on food products.

## 6. Biofilms and Bacteriophages

An important contributing factor to contamination in the food industry is the presence of biofilms on processing materials. Bacterial biofilms are communities of bacteria that have attached to a surface and formed an extracellular polymeric surrounding that renders them resistant to typical cleaning and disinfecting processes. Bacteria within biofilms develop this resistance due to differential growth and genetic expression than planktonic bacteria [112]. The persistence of these biofilms, especially in the food production industry has presented a need for new and novel antibacterial agents. Bacteriophages and phage derived lytic proteins have recently shown potential as antibacterial agents against biofilms [15,104,105,106,107,108,109,110,112].

In nature most bacteria live in naturally occurring biofilms. Considering this, bacteriophages naturally have developed the ability to infect bacteria in biofilms making them an ideal candidate for an antimicrobial against biofilms. Bacteriophages that target biofilms contain polysaccharide depolymerases that specifically hydrolyze polysaccharides and polysaccharide derivatives that make up the outer membrane of biofilms. The activity of these enzymes typically relies on the tailspike proteins of bacteriophages, a structure that serves to carry phage infection. The presence of this enzyme confers phages a significant advantage to other antimicrobial agents. Several studies have examined the effectiveness of bacteriophages against several species of biofilms formed on common food production surfaces including polystyrene, glass, and stainless steel [15,104,105,106,107,108,109,110,112] (Table 3). These studies exhibit the effectiveness of phage enzyme, mono-phage, and phage-cocktail treatments against a variety of biofilms.

## 7. Challenges of Bacteriophage Biocontrol

Research on bacteriophages as biocontrol agents for targeting pathogenic bacteria with high specificity has become increasingly prevalent. However, the advancement of phage biocontrol faces two major challenges: firstly, general consumer acceptance and secondly, resolution of technical constraints involved in development and standardization. Before phage preparations can be introduced as a replacement to standard treatment methods these two challenges must be addressed and resolved.

### 7.1. Consistent Efficacy

The most significant technical challenge of phage biocontrol is consistent efficacy. While in vitro and in vivo studies have illustrated success, the efficacy of phage treatments in commercial usage is highly dependent on the phages that are utilized within the treatment due to the associated specificity of phages. A common phenomenon observed with phage treatments is the initial decrease of bacterial colonization, followed by a restoration of bacterial growth [113,114]. The result of this is a drastic decrease of bacteria within the food source initially but not a complete elimination of the pathogen. Phage treatments must also be targeted to the location of colonization within the organism, leading some broad treatment methods to be ineffective in complete eradication. Therefore, oral administration of phages is typically successful in the treatment of intestinal tract infections but non-efficacious to secondary infections such as Colibacillosis-related meningitis infections [88]. Dosage determination is another technical challenge of phage treatments. Unlike antibiotics, phages are living organisms that can actively replicate within the body while lysing target bacteria. The number of phages as well as the conditions they are subject to within the body leads to variation in the rate of replication and production of viral progeny. This variation makes determination of exact dosage requirements challenging and requires further study. Theoretically phage concentration should increase exponentially as lysis of the target bacteria occurs, however recent studies have suggested that phages do not increase exponentially after application to food products [86,102,115]. This lack of exponential growth is related to the environmental conditions that typically have low water activity (aw) levels for preservation purposes. This low aw directly correlates to the reduction in phage mobility within the environment [34]. This obstacle could be avoided through increased phage concentrations which increases the probability of phage particles meeting the pathogen in food products with lower aw levels [68,116,117,118,119]. Another potential methodology to increase the efficacy of phage therapy is the use of high titer aerosol treatments. This method would be utilized following the harvesting of animals for food products and applied directly to the entire surface of the food product. This treatment could drastically reduce the contamination of food products, especially in cases such as *Campylobacter* contamination, where infective doses are relatively low. Standardization of proper application will ultimately resolve the challenge of phage efficacy in food product biocontrol.

### 7.2. Resistance Development

The development of resistance is another technical issue of phage biocontrol [72]. While bacteria have been shown to rapidly develop resistance to phage treatments in vitro, resistance has been only a minor problem for the effectiveness of phage therapy in real world applications [43]. The ability of phages to co-evolve with bacterial species to retain infectivity is a major factor that leads experts to have minimal concern regarding the long-term efficacy of phage-based treatments [43]. Bacterial host resistance to phages is primarily conferred through alteration of the structural components of the membrane. While these adaptations can limit the effectiveness of phages, it is important to note that they typically come at a cost for the bacteria. Development of phage-resistance by bacteria is typically dis-advantageous, resulting in a lack of surface features responsible for bacterial virulence [43,44,45,46]. The abundance of phages also presents a constant availability of distinct phages for use against various bacteria.

### 7.3. Public Acceptance

A challenge that addresses both the efficacy and public acceptance of phage biocontrol is the quantitative bacterial reduction associated with this treatment method. As outlined in previous sections, phage treatments typically reduce bacterial colonization of food products by approximately 1–3 log10 CFU [77]. This reduction is considerably lower than some hasher intervention methods such as chemical agents and irradiation, that reduce bacteria by approximately 5 log10 CFU [77]. This differentiation in CFU reduction may lead consumers and industry to shy away from phage therapy methods, however, there still is value in the reduction potential of this method. While the bacteria may not be fully eliminated from the food product it is still drastically reduced rendering it safer for consumption. Additionally, food products exhibiting log10 CFU levels greater than 3 are typically the result of other contamination issues introduced throughout the production process [120]. A study that exhibits the significance of even a 1–3 log10 CFU reduction is a risk assessment conducted jointly by the FDA and USDA’s FSIS that quantified the effect of contaminated deli meats on the morality rate of the elderly [121]. According to the analysis in this study, even a 10-fold (1 log10) and 100-fold (2 log10) reduction of bacteria in pre-retail contamination would reduce the mortality rate by approximately 50% and 74% respectively [121]. This data suggests that even though phage biocontrol methods do not always fully eradicate bacterial contamination they still may yield significant potential in applications of food safety and public health.

The final challenge for the introduction of phage biocontrol methods is public acceptance of the method. Recent consumer trends in terms of food appear to be aligning to provide entry of phage treated products into the market. Trends indicate that consumers are more likely to avoid chemically and antibiotically treated products in comparison to organic and locally produced ‘green’ products [7,8]. This trend is promising to phage biocontrol in that phage treatments are a natural, organic, and targeted approach to provide safe food products. While this is promising there is still a natural apprehension of consumers to the concept of using viruses on the food that they consume. This is mostly due to unfamiliarity with the method and the generally low scientific literacy of the common public. Before widespread utilization of this technology the public must be educated on the safety, efficacy, and ubiquity of bacteriophages in their food products.

## 8. Approved Phage Biocontrol Products

While the widespread acceptance of phages as biocontrol agents is still pending, some phage biocontrol products are currently approved and have begun being utilized in the food production industry. Most of these products are utilized in the Eastern World, where phage applications have been more exhaustively studied and are more widely accepted [122]. These commercial phage products are viewed as natural and green technologies. These phage products contain naturally occurring, GMO free phages and are typically suspended in water-based solutions that are free of harsh chemicals typically associated with biocontrol. Most importantly, these phage products, compared to other safety intervention methods that typically cost 10–30 cents per pound of treated food, only cost 1–4 cents per pound of treated food [40]. This drastic financial difference is another major advantage of phage biocontrol products in comparison to traditional intervention. It is important to note however that this is the cost associated with a product designed for very specific treatment of a singular pathogen. Incorporation of multiple products does increase the overall cost of phage biocontrol. Overall, the biological properties of lytic phages used in commercial phage biocontrol products make them a very attractive approach for further improvement of our food products. An increasing number of companies involved in the production and commercialization of phage biocontrol methods will further advance the relevance of this modality in the food production industry (Table 4).

## 9. Conclusions

The emergence of antibiotic resistant pathogens is a major threat to public health, especially when these pathogens are present in our food sources. The lack of antibiotic development and the continual rise of antibiotic resistant pathogens makes development of alternative antimicrobials essential to the preservation of public health and the prevention of foodborne outbreaks. While phage therapy is highly variable and dependent on target pathogen, location of infection, and complexity of the infection, recent studies outlined in this review demonstrate how phages can effectively remove these pathogens. The reduction of dangerous pathogens such as *Salmonella* spp., *E. coli*, *Campylobacter* spp., *Clostridium* spp., and *Listeria* spp. has a beneficial effect on both livestock and human health. Mitigation of these infections in livestock directly correlates to the safety of the food products that we consume [3]. Overall, inclusion and further development of phage products not only would increase the safety of food products but also decrease the associated economic burden of foodborne pathogens on the food industry. As expressed throughout this review, poultry production is the industry in which we could first see widespread application of phage biocontrol [33,68,69,70,71,83]. Unlike some food production processes, the poultry industry utilizes highly integrated production systems that are more amenable to phage therapy. Poultry production systems allows for the flexibility to introduce phages at multiple points in production, including feed, water, aerosol sprays, and modified packaging [68,69,70,71,82,83]. Before these treatments are utilized it is key for the installation of a regulatory framework that would allow for the incorporation of phage treatments. Organizations such as the USDA in the United States or the EU in Europe are responsible for this framework.

A potential drawback of phage products is the ability of bacterial species to develop resistance mechanisms against phages [43,133]. While resistance development is an obstacle of any treatment method it is important to note that resistance to one phage does not confer resistance to all phages. For this reason, phage cocktails appear to be an ideal administration method for biocontrol [68,71,75,86,87]. Literature also suggests there is a fitness cost associated with the development of phage resistance in bacteria resulting decreased growth of bacteria even without the presence of phage [135]. Another aspect of phage therapy that must be considered is the specificity of phages and what this means in terms of intervention design. When considering the host specificity of phage, it may be beneficial to target bacterial species that are genetically homologous. Genetically homologous pathogens are more attractive targets for phage therapy in that there is not a wide variation phenotypical and genotypical factors that must be taken into consideration. For instance, bacterial species such as *E. coli* pose difficulties for phage treatments due to their wide genetic diversity. Finally, it is important that phage intervention is used responsibly to prolong the efficacy of this method and avoid the issues that we now face with modern day antibiotic intervention methods. Potential methods that could circumvent issues associated with antibiotic treatments would be regulatory rotation or reformulation of phages used in intervention methods to retain efficacy and avoid resistance. Overall, the future of phages in the food production industry is promising and quickly approaching. As discussed throughout this review there are now numerous approved phage treatments designed for use in food production [73,117,118,127,128,129,130,131,132,133]. The continued incorporation of phages into food production should encourage research and development of these products.

## Figures and Tables

**Figure 1 microorganisms-10-02126-f001:**
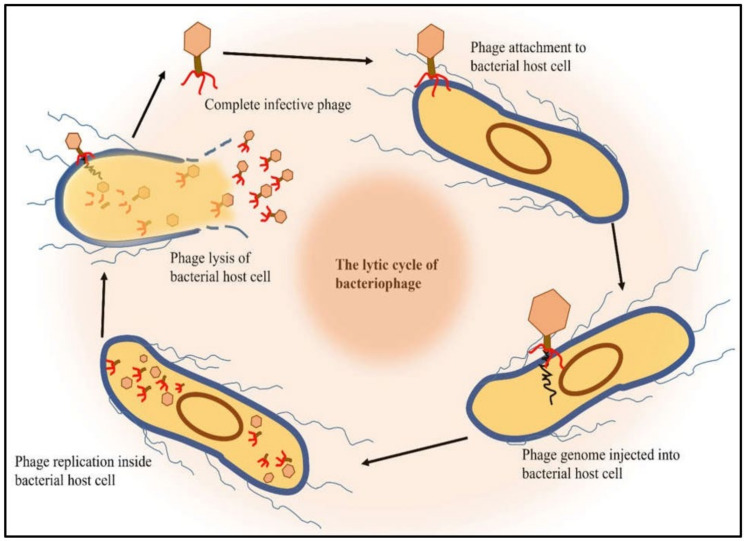
Lytic cycle of bacteriophages. The lytic cycle is one of the two cycles of bacteriophage replication that leads to the lysis and destruction of the infected cell. These bacteriophages that employ the lytic cycle have been shown to have significant importance in biocontrol of bacteria in livestock infections, where they target bacteria with high host-specificity. In the lytic cycle, the bacteriophage genome does not integrate with the host bacterial cell genome, instead, the DNA is found in the cytoplasm of the host cell as separate and free-floating molecule, replicating separately from the host bacterial DNA.

**Figure 2 microorganisms-10-02126-f002:**
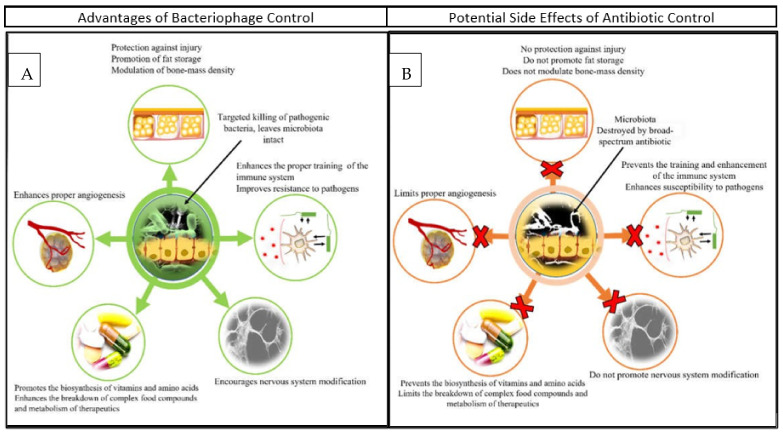
Comparison of bacteriophage (**A**) and antibiotic (**B**) application and their respective impact on livestock.

**Table 1 microorganisms-10-02126-t001:** General Advantages (Top) and Disadvantages (Bottom) of Bacteriophages as Biocontrol Agents.

Advantages of Bacteriophage Biocontrol		
Naturally Abundant	Bacteriophages are the most abundant organisms in the biosphere. This abundance makes them readily available for isolation and use.	[25]
High Specificity	Bacteriophages infect their hosts with high specificity allowing for preservation of natural microbiota and ensuring elimination of only the target bacteria.	[17,37]
Rapidly Produces Viral Progeny	Unlike modern therapeutics, such as antibiotics, bacteriophages feature the ability to reproduce once they encounter and infect their target bacteria.	[38]
Versatile Application	The ability to genetically and chemically modify phages paired with their structural flexibility, cargo capacity, ease of propagation, and generally safe characteristics provide a variety of potential avenues of application.	[39]
Harmless to Natural Microbiota	Bacteriophages high specificity to their host allows these viruses to only carry out lysis on target bacteria. This preserves natural microbiota that cannot be infected by these bacteriophages.	[17,19,24]
Cost-Effective	The abundance and stability of bacteriophages allow them to be easily isolated, purified, and reproduced. Bacteriophage treatment typically costs only 1–4 cents compared to traditional intervention costing approximately 10–30 cents.	[40]
Active Against MDR Bacteria	Bacteriophages and their ability to co-evolve with bacterial species allows for these viruses to retain efficacy even against MDR bacterial species.	[41,42]
**Disadvantages of Bacteriophage Biocontrol**		
Bacterial Resistance	The ability of bacteria to develop resistance to bacteriophages has been described throughout literature. However, the ability of phages to co-evolve with bacterial species allows for retention of efficacy. This co-evolution paired with the abundance of available phages makes resistance a minimal concern.	[43,44,45,46]
Immunogenicity	Bacteriophages are non-self-antigens that can be recognized by the immune system. The effectiveness of phages relies highly on phage strain, route of administration, and prior exposure. In some cases, the immune system actively recognizes and destroys bacteriophages.	[47,48]
Narrow Host Range	Bacteriophage host specificity while beneficial to natural host microbiota, can lead to issues of efficacy. The specificity of phages requires isolation and characterization of target pathogens for proper phage selection prior to use. This characterization introduces hurdles into the application of phage therapy.	[49]
Regulatory Approval	The unconventional aspects of phage therapy and their lack of conformity to the canonical model have impacted the development of suitable regulatory pathways. This issue is seen commonly in the Western world.	[50]
Lack of Well-Define Delivery Systems	Numerous delivery systems have been described in the literature including oral, intramuscular, intercranial, and intravenous administration. The success of these delivery systems varies based on infection type and location. For the approval and incorporation of bacteriophage therapy it is important to standardize delivery systems for the progression of phage approval.	[49]

**Table 2 microorganisms-10-02126-t002:** Phage Biocontrol Studies. The studies referenced in Table 2 outline several methods in which phages can be utilized for the prevention and control of *Salmonella*, *E. coli*, *Campylobacter*, and *Clostridium* within food sources [68,69,70]. The versatility of phage therapy as a preventative treatment as well as a therapeutic is a benefit not possessed by typical chemical intervention methods.

Phage Treatment	Target	Results	Reference
**Lytic Phage Cocktail**	*S. enteritidis*, *S. hadar*, *S. typhimurium*	*Salmonella* colonization of the ceca was reduced approximately 2.19–2.53 log10 CFU/g.	[33]
**Mono-phage Treatment as a** **feed additive**	Prevention of Horizontal infection by *S. enteritidis*	Significant reduction (*p* < 0.05) of intestinal colonization by up to 1 log10 CFU/g after 21 days and prevention of *S. enteritidis* infection through horizontal infection.	[69]
** *Salmonella* ** **Specific Phage** **Cocktail incorporated with** **water supply**	Reduction of *Salmonella* colonization in the intestine	Reduction of mean *Salmonella* colonization of the Intestinal tract by up to 1.6 log10 CFU/mL.	[71]
**Prophylactic Mono-phage** **Treatment (PSE)**	*S. enteritidis* within the ceca	Prophylactic Treatment reduced the detection of *S. enteritidis* to 20% 7 days after administration.	[70]
**Six-phage Cocktail**	*S. Paratyphi*-B S661 Various *Salmonella* spp.	Reduction of all *Salmonella* population ≈ 2–4 log_10_ on all surfaces	[72]
**Mono-phage Treatment**	*Salmonella Enteritis* (*SeE Nal^r^*)	Slight reduction of SeE Nal^r^ in the cacum of chickens but not significantly different	[73,74]
**High Titer Phage Cocktail Aerosol**	*E. coli* spp.	Reduction of mortality rate of *E. coli* challenged chickens	[75]
**Direct Intramuscular** **Injection** **(DAF6 and SPR02)**	*E. coli* spp.	Reduction of mortality rate and effective even 48 h after symptoms	[76]
**Phage Cocktail Inoculation** **(CP14 and CP68)**	*C. jejuni*	3 log_10_ reduction of *C. jejuni* in Vrolix chicken caeca	[77]
**Oral Phage Cocktail** **(CP14 and CP68)**	*C. jejuni*	2 log_10_ reduction of *C. jejuni* in broiler chicken caeca	[73,78,79]
**Phage Endolysins**	*C. perfringens*	Induced bacterial lysis in infected chickens	[80]
**Phage Cocktail** **INT-401**	*C. perfringens*	Reduced C. perfringens-related mortality by 92% in chickens	[81]

**Table 3 microorganisms-10-02126-t003:** Brief overview of bacteriophages against biofilm.

Studies of Bacteriophages Against Biofilms			
Phage/Phage Protein	Target Bacteria	Efficacy of Treatment	Reference
Phages LiMN4L, LiMN4p, LiMN17	*L. monocytogenes*	Phage cocktail reduced biofilm on stainless steel after 75 min.	[104]
Phage P100	*L. monocytogenes*	Mono-phage treatment reduced biofilm on stainless steel to undetectable after 48 h.	[105]
Phages CP8 and CP30	*C. jejuni*	Phage cocktail treatment reduced biofilm on glass 1–3 log units/cm^2^	[106]
Phage KH1	*E. coli* O157:H7	Reduction of 1.2 log units per coupon after 4 days of treatment at 4 °C	[107]
BEC8 Cocktail	*E. coli* O157:H7	Reduction of biofilm on stainless steel, ceramic tile, and polyethylene after 1 h of treatment at 37, 23, and 12 °C.	[108]
Phage T4	*E. coli* O157:H7	Cotreatment of polystyrene with T4 and cefotaxime resulted in complete elimination of biomass.	[109]
Endolysin Lys68	*S. typhimurium*	Phage protein treatment reduced biofilm cell counts by 1 log unit after 2 h of treatment in the presence of membrane permeabilizers	[110]
Epsilon 34 Tail Spike Protein	*S. newington*	Phage protein treatment reduced biofilm formation in an ex vivo cartilage model.	[15]
Four-phage cocktail	*C. difficilie*	Phage cocktail effectively reduced biofilm in animal model (*G. mellonella*).	[111]

**Table 4 microorganisms-10-02126-t004:** Commercial Phage Biocontrol Products. Adapted and modified from Moye et al., 2018, A brief list of relevant commercial phage biocontrol products.

Company	Phage Product	Target Organism(s)	Regulatory	References
**FINK TEC GmbH** **(Hamm, Germany)**	Secure Shield E1	*E. coli*	FDA, GRN	[72,117,123,124,125,126,127]
	Ecolicide^®^ (EcolicidePX™)	*E. coli* *O157:H7*	USDA, FSIS	
	EcoShield™	*E. coli* *O157:H7*	FDA, FCN 1018; Israel Ministry of Health; Health Canada	
**Intralytix, Inc.** **(Baltimore, MD, USA)**	SalmoFresh™	*Salmonella* spp.	FDA, GRN 435, USDA, FSIS Directive 7120.1; Israel Ministry of Health; Health Canada	[128,129]
**Micreos Food Safety (Wageningen, Netherlands)**	PhageGuard S™	*Salmonella* spp.	FDA, GRN 468; FSANZ; Swiss BAG; Israel Ministry of Health; Health Canada	[130,131]
		*E. coli* *O157:H7*	FDA, GRN 757 (Pending)	
**Passport Food Safety Solutions (West Des Moines, IA, USA)**	Finalyse^®^	*E. coli* *O157:H7*	USDA, FSIS Directive 7120.1	[132]
**PhageLux** **(Shanghai, China)**	SalmoPro^®^	*Salmonella* spp.	FDA, GRN 603	[133]
**Intralytix, Inc.** **(Baltimore, MD, USA)**	ListShield™	*L. monocytogenes*	FDA, 21 CFR 172.785; FDA, GRN 528; EPA Reg No. 74234-1; Israel Ministry of Health; Health Canada	[73,118]
**Intralytix, Inc.** **(Baltimore, MD, USA)**	PhageGuard Listrex™	*L. monocytogenes*	FDA, GRN 198/218; FSANZ; EFSA; Swiss BAG; Israel Ministry of Health; Health Canada	[73,134]

## Data Availability

Data is contained within the article.

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
