# Peer review of "Bacteriophages as Biocontrol Agents in Livestock Food Production"

_microorganisms, 2022, doi:10.3390/microorganisms10112126_

Round 1

Reviewer 1 Report

Green biocontrol options are important to prevent and mitigate infectious disease in food production. Bacteriophages have great potential as biocontrol agents in food production. This review briefly introduces phage biocontrol, its application in food production, its efficacy against foodborne pathogens including Salmonella, E. coli, Campylobacter and Clostridium, the challenges involved in development, and current approved products. The manuscript could be published after proper revision.

Line 27 to 28, “The most efficient antimicrobial methods….and the utilization of antimicrobial agents…..”. Please tone down. The utilization of antimicrobial agents is based on what kind of food products

Line 51, “…in numerous food and beverage products”. Please list a few examples.

For the section of “the role of bacteriophages in food microbiology”, please consider having a table summarizing the general advantages and disadvantages of bacteriophages as biocontrol agents to provide a comprehensive comparison.

Table 1, Salmonella, italic.

The title of this study seems to talk about application of bacteriophages in multiple food production. However, only the utilization in livestock is covered. Bacteriophages as biocontrol agents for food safety could be used in livestock food production, agricultural food production aquaculture and fish food production. Please consider modifying the title or improving the context.

This study includes the utilization of bacteriophages in bacterial pathogens control, the authors may also consider its application on the control of bacterial biofilms.

For the challenges, the authors may want to list or summarize in a table or under subtitle to be clearer. Also, any concerns related to bacterial resistance to phages?

Author Response

Green biocontrol options are important to prevent and mitigate infectious disease in food production. Bacteriophages have great potential as biocontrol agents in food production. This review briefly introduces phage biocontrol, its application in food production, its efficacy against foodborne pathogens including SalmonellaE. coliCampylobacter and Clostridium, the challenges involved in development, and current approved products. The manuscript could be published after proper revision.

Line 27 to 28, “The most efficient antimicrobial methods….and the utilization of antimicrobial agents…..”. Please tone down. The utilization of antimicrobial agents is based on what kind of food products

Language has been toned down. Utilization on basis of food product type has been added.

Line 51, “…in numerous food and beverage products”. Please list a few examples.

Examples are not listed. 

For the section of “the role of bacteriophages in food microbiology”, please consider having a table summarizing the general advantages and disadvantages of bacteriophages as biocontrol agents to provide a comprehensive comparison.

Table listing and describing general advantages and disadvantages has been added to provide comprehensive comparison. 

Table 1, Salmonella, italic.

Salmonella is now italicized. 

The title of this study seems to talk about application of bacteriophages in multiple food production. However, only the utilization in livestock is covered. Bacteriophages as biocontrol agents for food safety could be used in livestock food production, agricultural food production aquaculture and fish food production. Please consider modifying the title or improving the context.

Title altered to Bacteriophages as Biocontrol Agents in Livestock Food Production to better fit content of the review. 

This study includes the utilization of bacteriophages in bacterial pathogens control, the authors may also consider its application on the control of bacterial biofilms.

Section covering biofilms and listing studies concerning biofilm formation in food production and treatment with bacteriophages added. 

For the challenges, the authors may want to list or summarize in a table or under subtitle to be clearer. Also, any concerns related to bacterial resistance to phages?

Subsection titles have been added to distinguish challenges discussed. Section on bacterial resistance has been added to challenges section. 

Reviewer 2 Report

The review "bacteriophages as Biocontrol Agents in Food Microbiology" shows a synthesis of the applications of different solutions in the industry 

However, there are some topics that could be improved:

Line 51-52: I suggest focusing on the use of bacteriophages and their derivatives instead of talking about other types of biocontrollers.

I consider that figures 2 and 3 could be consolidated into only one figure and that the information contained in them should be mentioned and described in the text.

There are more examples that can be used in Table 1. Example: Viruses 2018, 10(4), 205; https://doi.org/10.3390/v10040205

Examples of the other bacteria described in the review (E. coli, Campylobacter, Clostridium) could be added to this table.

The review develops examples of phage-based biocontrollers for Salmonella, E. coli, Campylobacter and Clostridium, but leaves others out, such as Listeria, that must be incorporated.

Table 2 is an extract from the cited table, it could be supplemented with other current information (https://doi.org/10.1111/j.1472-765X.2008.02458.x)

Among the disadvantages of the use of cocktails of bacteriophages, the phenomenon of resistance must be developed.

The conclusions must be consistent with the development of the article

Author Response

The review "bacteriophages as Biocontrol Agents in Food Microbiology" shows a synthesis of the applications of different solutions in the industry 

However, there are some topics that could be improved:

Line 51-52: I suggest focusing on the use of bacteriophages and their derivatives instead of talking about other types of biocontrollers.

Section has been altered to focus more on bacteriophages and their derivatives. Extensive information on other biocontrollers has been removed. 

I consider that figures 2 and 3 could be consolidated into only one figure and that the information contained in them should be mentioned and described in the text.

Figures have been consolidated into one. Information has been added to the section and removed from caption. Figure caption has been augmented. 

There are more examples that can be used in Table 1. Example: Viruses 2018, 10(4), 205; https://doi.org/10.3390/v10040205

More examples that are discussed within the manuscript have been added to Table 1 from mentioned publication. Table 1 is now Table 2.

Examples of the other bacteria described in the review (E. coli, Campylobacter, Clostridium) could be added to this table.

Table moved to end of section and is now comprehensive of all bacterial species discussed within the section. 

The review develops examples of phage-based biocontrollers for Salmonella, E. coli, Campylobacter and Clostridium, but leaves others out, such as Listeria, that must be incorporated.

Section added to discuss Listeria and its impact in food production as well as relevant bacteriophage-Listeria studies. 

Table 2 is an extract from the cited table, it could be supplemented with other current information (https://doi.org/10.1111/j.1472-765X.2008.02458.x)

Table 2 is now Table 4. Using the reference publication more current information was added to the table. 

Among the disadvantages of the use of cocktails of bacteriophages, the phenomenon of resistance must be developed.

Section discussing bacterial resistance has been added to challenges section. 

The conclusions must be consistent with the development of the article

Conclusion has been altered for consistency. 

Round 2

Reviewer 2 Report

The review was corrected and improved. Just is necessary to incorporate some detail :

Line 22-23: …gens such as Salmonella spp., Escherichia coli (E. coli), Campylobacter spp., and Clostridium spp. [1,2]. Write  the microrganismo by italics

Figure1: Recommendation: draw the genetic material of the bacteriophage in the stage replication of the phage in the bacterila host cell

Lines 109-110: At the ends of phase, add references :

ous other common foodborne bacteria have been accepted by the FDA as safe additives to food products.

The authors must to cite Table N1

The authors must to use italics in the table N2 for the microganisms name.